# Immune Cell Deconvolution Reveals Possible Association of γδ T Cells with Poor Survival in Head and Neck Squamous Cell Carcinoma

**DOI:** 10.3390/cancers15194855

**Published:** 2023-10-05

**Authors:** Anuraag S. Parikh, Yize Li, Angela Mazul, Victoria X. Yu, Wade Thorstad, Jason Rich, Randal C. Paniello, Salvatore M. Caruana, Scott H. Troob, Ryan S. Jackson, Patrik Pipkorn, Paul Zolkind, Zongtai Qi, Douglas Adkins, Li Ding, Sidharth V. Puram

**Affiliations:** 1Department of Otolaryngology—Head and Neck Surgery, Columbia University Irving Medical Center, New York, NY 10032, USA; vxy2101@cumc.columbia.edu (V.X.Y.); sc2876@cumc.columbia.edu (S.M.C.); sht22@cumc.columbia.edu (S.H.T.); 2Herbert Irving Comprehensive Cancer Center, Columbia University Irving Medical Center, New York, NY 10032, USA; 3Department of Medicine, Division of Oncology, Washington University in St. Louis, St. Louis, MO 63110, USAdadkins@wustl.edu (D.A.); lding@wustl.edu (L.D.); 4McDonnell Genome Institute, Washington University in St. Louis, St. Louis, MO 63108, USA; 5Department of Otolaryngology—Head and Neck Surgery, Washington University School of Medicine, St. Louis, MO 63108, USA; amazul@wustl.edu (A.M.); richj@wustl.edu (J.R.); paniellor@wustl.edu (R.C.P.); jackson.ryan@wustl.edu (R.S.J.); ppipkorn@wustl.edu (P.P.); pzolkind@wustl.edu (P.Z.); qizongtai@gmail.com (Z.Q.); 6Department of Radiation Oncology, Washington University in St. Louis, St. Louis, MO 63110, USA; thorstad@wustl.edu; 7Siteman Cancer Center, Washington University in St. Louis, St. Louis, MO 63108, USA; 8Department of Genetics, Washington University School of Medicine, St. Louis, MO 63108, USA

**Keywords:** head and neck squamous cell carcinoma, deconvolution, immune microenvironment, gamma−delta T cells, survival

## Abstract

**Simple Summary:**

In head and neck squamous cell carcinoma (HNSCC), a major challenge with treatments targeting the immune system is an incomplete understanding of rare immune cell subtypes. Several recent studies have explored these cell types, but these studies have not integrated cellular data with clinicopathologic or demographic features to place conclusions in an appropriate clinical context. We deconvolve transcriptomic data from HNSCC patients to infer proportions of immune cell subtypes and place these data in the context of demographic, clinical, pathologic, and genomic characteristics. Our analysis revealed a possible association of γδ T cells with poor survival in HNSCC and underscore the need to better understand the role of these rare cells in HNSCC, including whether the presence of γδ T cells may predict the need for adjuvant therapy.

**Abstract:**

(1) Background: The role of rare immune cell subtypes in many solid tumors, chief among them head and neck squamous cell carcinoma (HNSCC), has not been well defined. The objective of this study was to assess the association between proportions of common and rare immune cell subtypes and survival outcomes in HNSCC. (2) Methods: In this cohort study, we utilized a deconvolution approach based on the CIBERSORT algorithm and the LM22 signature matrix to infer proportions of immune cell subtypes from 517 patients with untreated HPV-negative HNSCC from The Cancer Genome Atlas. We performed univariate and multivariable survival analysis, integrating immune cell proportions with clinical, pathologic, and genomic data. (3) Results: We reliably deconvolved 22 immune cell subtypes in most patients and found that the most common immune cell types were M0 macrophages, M2 macrophages, and memory resting CD4 T cells. In the multivariable analysis, we identified advanced N stage and the presence of γδ T cells as independently predictive of poorer survival. (4) Conclusions: We uncovered that γδ T cells in the tumor microenvironment were a negative predictor of survival among patients with untreated HNSCC. Our findings underscore the need to better understand the role of γδ T cells in HNSCC, including potential pro-tumorigenic mechanisms, and whether their presence may predict the need for alternative therapy approaches.

## 1. Introduction

Immunotherapies have demonstrated durable survival benefits in a small fraction of patients with recurrent and metastatic head and neck squamous cell carcinoma (HNSCC) [1,2]. A major challenge with their widespread utility in HNSCC has been an inability to reliably predict response. One factor contributing to this limitation is an incomplete understanding of the composition and roles of immune cell subtypes in the HNSCC tumor microenvironment (TME) [3,4]. Immunotherapies targeting tumor infiltrating lymphocytes (TIL) have spurred investigations into the proportions and functional states of T lymphocytes, specifically [5,6,7]. Accordingly, some aspects of TIL in HNSCC have been elucidated, including the importance of cytotoxic and regulatory T cells and the concept of T cell exhaustion. However, the roles of rare immune cell subsets remain less clear.

With the large quantity of publicly available bulk transcriptomic data, several recent studies have utilized computational deconvolution efforts to explore immune cell subtypes [8,9,10,11,12,13]. However, despite describing similar methodologies for deconvolution, these studies derived disparate conclusions regarding the prognostic associations of the immune infiltrate, overall [8,9,10], as well as specific immune cell subtypes [8,11,12]. Most importantly, existing studies have not integrated novel data on immune cell subtypes with demographic, clinicopathologic, or genomic features, including those with known clinical utility, to validate conclusions or to place them in an appropriate clinical context.

Here, we utilize the cell-type identification by estimating relative subsets of RNA transcripts (CIBERSORT) algorithm and the LM22 signature matrix [14] to deconvolve HNSCC bulk RNA-sequencing data from The Cancer Genome Atlas (TCGA) and integrate it with clinicopathologic and genomic data in a multivariable survival analysis to examine the independent associations of immune cell subtypes with survival in HPV-negative HNSCC.

## 2. Materials and Methods

***Bulk RNA-Seq data and clinical information of HNSCC tumors from TCGA.*** Bulk RNA-sequencing data of HNSCC tumors were obtained from The Cancer Genome Atlas (TCGA) database (https://portal.gdc.cancer.gov/, dbGaP study accession phs000178, accessed on 1 September 2019). A total of 566 cases were selected using the following filtering criteria: primary site = head and neck, disease type = squamous cell neoplasms, sample type = primary tumor, experimental strategy = RNA-seq, and workflow type = HTseq (High Throughput sequencing) counts or FPKM (Fragments Per Kilobase Million). To maintain the biological uniformity of the cohort, HPV-positive oropharynx HNSCC patients were eliminated, with a filtered dataset of 517 cases. Clinical information for patients with HNSCC, including age, gender, smoking status, T stage, N stage, perineural invasion, lymphovascular invasion, and survival in months, was accessed on 1 September 2019. 

***CIBERSORT deconvolution analysis*.** We utilized the CIBERSORT online tool implementation [14] to infer the proportions of 22 immune cell subtypes from bulk RNA-seq data. The HTseq-FPKM normalized expression data for 517 HPV-negative HNSCC cases from TCGA and the LM22 signature matrix were used as the inputs for CIBERSORT, as previously described [14]. The LM22 signature matrix contains transcriptional signatures for 22 immune cell subtypes, including naïve B cells, memory B cells, plasma cells, CD8 T cells, naïve CD4 T cells, memory resting CD4 T cells, memory activated CD4 T cells, follicular helper T cells, regulatory T cells, γδ T cells, resting natural killer (NK) cells, activated NK cells, monocytes, M0 macrophages, M1 macrophages, M2 macrophages, resting dendritic cells, activated dendritic cells, resting mast cells, activated mast cells, eosinophils, and neutrophils. A CIBERSORT *p*-value cutoff of 0.05 was used to filter out cases with an unreliable deconvolution, leaving a final dataset of 317 HPV-negative HNSCC cases. This analysis provided sample-level fractions of 22 individual immune cell subtypes. Average cell type fractions across samples were computed, as well as Pearson correlations between cell types.

***Survival analysis*.** A univariate Cox proportional hazards model was used to assess the impact of 22 immune cell proportions, as computed by CIBERSORT, on disease-free survival (DFS). Disease-free survival was defined as recurrence or death, whichever was first. For this analysis, tumors were categorized as either subtype^high^ (top quartile) or subtype^low^ (bottom three quartiles) for each immune cell subtype. For cell types that were detected in less than 25% of tumors, those tumors in which the cell type was detected (subtype = 1) were compared with those in which it was not detected (subtype = 0). Kaplan−Meier survival curves were generated for DFS when stratified by immune cell subtypes found to be significant in the univariate analysis, as well as other immune cell subtypes of potential prognostic significance. Immune cell subtypes with significant associations in the univariate analysis were included in the multivariable Cox proportional hazards model, along with additional demographic, clinical, pathologic, and genomic features, including age (<60 vs. 60+), smoking status (never/former vs. current), T stage (T1-T2 vs. T3–T4), N stage (N0–N1 vs. N2–N3), subsite (oral cavity, larynx, or HPV-negative oropharynx), and TCGA expression subtype (basal, atypical, classical, or mesenchymal). Parameters with significant missing values, including perineural invasion (PNI) and lymphovascular invasion (LVI), were eliminated from the multivariable analysis due to concern for critical reduction in sample size. To compare systematic differences in clinicopathologic parameters between patients with and without γδ T cells, chi-squared tests were performed. All statistical analyses were performed using R (4.2.0) and a *p*-value cutoff of 0.05.

***Differential gene expression and pathway analysis*.** Gene expression was used to perform pairwise differential analysis between groups of samples. A Wilcoxon rank-sum test was performed to determine the differential expressed genes (DEG). The *p*-value was adjusted using the Benjamini−Hochberg procedure, and genes were considered significant with an adjusted *p*-value < 0.05. For each comparison, we obtained the top DEGs ranked by the highest fold change that were significantly different between the comparison groups (FDR < 0.05). We used ConsensusPathDB-human for gene set over-representation analysis.

## 3. Results

***CIBERSORT deconvolution provides proportions of 22 immune cell subtypes.*** To infer the proportions of common and rare immune cell subtypes in HNSCC, we utilized the CIBERSORT algorithm and the LM22 immune cell signature matrix [14] to deconvolve bulk TCGA RNA-seq data for 517 HPV-negative HNSCC tumors (Figure 1A). Samples without reliable deconvolution (*p* > 0.05) were removed, and the final filtered dataset contained 317 tumors, including 226 oral cavity, 70 larynx, and 21 oropharynx HPV-negative tumors. Immune cell proportions varied widely across cell types and tumors. The most common immune cell types were M0 macrophages (25.6 ± 0.9%), M2 macrophages (13.0 ± 0.4%), and memory resting CD4 T cells (10.4 ± 0.4%), while the least common immune cell types were eosinophils (0.1 ± 0.0%), γδ T cells (0.1 ± 0.0%), and naïve CD4 T cells (0.1 ± 0.0%) (Figure 1B). To ensure that immune cell types were not highly correlated and, instead, represented independent features of the tumor immune microenvironment, we assessed Pearson correlations between each pair of immune cell types and found no significant correlations in our dataset (Figure 1C), thus supporting the validity of the 22-cell-type classification used.

***Univariate analysis reveals rare T cell subtypes associated with survival.*** To assess the association of immune cell subtypes with DFS, we first performed a univariate analysis using a Cox proportional hazards model (Appendix A). In this analysis, only two immune cell subtypes showed a significant association with survival: memory activated CD4 T cells (*p* = 0.02) and γδ T cells (*p* < 0.01). Kaplan−Meier curves for these two cell types, as well as other cell types of interest within HNSCC, were then generated (Figure 2). Interestingly, while a higher proportion of infiltrated CD4 memory T cells was associated with better DFS (Figure 2A), a non-zero fraction of γδ T cells was associated with a poorer DFS (Figure 2B). By contrast, proportions of CD8 T cells (Figure 2C), regulatory T cells (Figure 2D), activated M1 macrophages (Figure 2E), and activated NK cells (Figure 2F) had no association with DFS in our cohort.

***Multivariable analysis confirms independent association of**γ**δ T cells with survival.*** To determine whether immune cell subtypes were independently associated with DFS in our cohort, we then developed a rigorous multivariable Cox proportional hazards model, in which significant cell types from the univariate analysis were included, along with additional demographic, pathologic, and genomic features. Complete case analysis was performed, with a total of 287 patients included. As expected, higher N stage was highly significantly associated with poorer DFS (HR 1.70, 95% CI [1.18–2.45], *p* < 0.005, Table 1). Interestingly, the only other feature that was significantly associated with DFS was the presence of γδ T cells, which was associated with poorer DFS (HR 2.13, 95% CI [1.10–4.13], *p* < 0.05, Table 1). There was no association of age, smoking status, TCGA molecular subtype, anatomic site, T stage, or proportion of memory activated CD4 T cells with survival (Table 1). We then compared demographic and clinicopathologic features of patients with (n = 14) and without (n = 303) γδ T cells and found no significant associations with gender, smoking status, T stage, N stage, PNI, LVI, anatomic subsite, or TCGA subtype (Appendix A), confirming that the presence of γδ T cells is a truly independent negative predictor of survival.

Differential expression and pathway analysis fail to reveal a clear transcriptional signature associated with γδ T cell infiltration. Finally, to explore potential mechanisms behind the poor DFS associated with the presence of γδ T cells, we examined differential gene expression among patients with and without γδ T cells. Interestingly, several genes were significantly upregulated in patients without γδ T cells, but no genes upregulated in patients with γδ T cells (Appendix A). We then conducted a pathway analysis to determine if there were any coherent patterns in differential gene expression and found increased expression of pathways related to gene expression and transcription in tumors with γδ T cells (Appendix A).

## 4. Discussion

The advent of immunotherapy as a novel treatment modality has sparked a major interest in understanding the roles of immune cell subtypes in solid tumors. Here, we performed an immune cell deconvolution of TCGA bulk RNA-seq data for 517 untreated HPV-negative HNSCC tumors and uncovered a novel independent association of γδ T cells with poor DFS. Our study provides a rich and comprehensive analysis of the tumor immune microenvironment composition in HNSCC, which should be a valuable resource for future studies. Importantly, unlike prior studies, we combined immune cell deconvolution with clinicopathologic and genomic features in a multivariable analysis. These analyses revealed γδ T cells and N stage as independently associated with survival, uncovering a novel potential role for γδ T cells in HNSCC biology, while affirming the established role of nodal stage as an important adverse pathologic feature in HPV-negative HNSCC. Together, these findings represent a critical observation likely to inform future studies, as well as HNSCC prognostication and therapeutics.

γδ T cells are a subgroup of T lymphocytes with T cell receptors with gamma and delta chains [15]. γδ T cells are much less common than αβ T cells but recognize antigens in a non-MHC restricted fashion and have a high secretory capacity for cytokines, suggesting potential for antitumor efficacy [15]. Accordingly, γδ T cells have been proposed to be associated with improved outcomes in many solid tumors, sparking interest in the development of immunotherapies targeting this rare subset [16]. However, γδ T cells also secrete pro-angiogenic IL-17 and promote proliferation of myeloid derived suppressor cells, and thereby may have potential pro-tumorigenic functions [15]. Thus, while γδ T cells isolated from peripheral blood are cytotoxic to oral tumor cells in vitro [17] and γδ T cells, as estimated by TRDC, TRGC1, and TRGC2 expression in bulk RNA-seq data, were associated with improved prognosis in HNSCC [18], Bas et al. found an increased fraction of peripheral blood γδ T cells in patients with HNSCC relative to healthy controls, as well as higher fractions in patients with recurrent disease [19]. These conflicting results may be related to difficulty in accurately quantifying γδ T cells by gene expression or flow cytometry due to overlap in expression with NK cells and CD8 T cells [20]. They also underscore the importance of specifically understanding the prognostic role of γδ T cells in HNSCC, including any potential role in facilitating the progression from dysplasia to carcinoma.

In the present study, we found an independent association of γδ T cells with poor prognosis in HPV-negative HNSCC. Our use of CIBERSORT with the LM22 signature matrix enabled us to distinguish immune cell subtypes with overlap in expression and to accurately quantify rare immune cell subsets. Interestingly, only 14 patients (4.4%) had non-zero fractions of γδ T cells, underscoring the rarity of these cells. It has been estimated that approximately 1–10% of peripheral blood CD3 positive cells are γδ T cells [16,19]; in our dataset, γδ T cells comprised < 0.1% of all immune cells and 0.3% of all T cells. This discrepancy may be related to overestimation of γδ T cells when measured by flow cytometry or gene expression, or to differences in fraction between peripheral blood and tumor tissue.

The association of γδ T cells with decreased DFS was maintained in the multivariable analysis, which accounted for numerous other demographic, clinical, pathologic, and genomic factors, indicating that γδ T cells are an independent predictor of survival. Unlike prior studies, our work specifically incorporated these other well-established predictors into our model, thereby more robustly querying the role of γδ T cells in patient outcomes. Importantly, the adverse significance of nodal stage was also maintained in this analysis, further validating the clinical utility of the findings related to γδ T cells. Finally, this relationship was confirmed by the similarities in patient demographics and tumor features when comparing patients with and without γδ T cells.

The present work has several limitations. First, although the use of LM22 to deconvolve immune cell subtypes with overlapping expression is a strength of the present work, it is also a limitation, in that our ability to deconvolve cellular subsets is limited to the 22 cell types present in LM22. Accordingly, we were unable to deconvolve myeloid derived suppressor cells, previously proposed as a mechanism by which γδ T cells may be pro-tumorigenic [15]. With the lack of consensus in the literature on the significance of γδ T cells, further exploration into potential pro-tumorigenic mechanisms in head and neck cancer is critical. Moreover, 200 patients were lost due to insufficient deconvolution, thus limiting the sample size used in the final analyses. Second, given the rarity of γδ T cells, these cells were only detected in 14 patients in our cohort. As a result, these analyses may thus more accurately be seen as hypothesis-generating, and there is a need to reproduce these results in an independent cohort to support clinical utility. In addition, given the predominance of oral cavity subsite tumors in TCGA, these results warrant validation in larger cohorts of larynx and oropharynx SCC patients to determine whether these results extend to other subsites. Moreover, with the number of clinical and immune cell variables investigated, even in a cohort as large as ours and with the use of a multivariable analysis, the possibility of a false positive result remains, underscoring the need for prospective validation. Third, as treatment paradigms of patients included in TCGA may vary and are not detailed explicitly, cross-institutional and subsite-related differences may drive differences in survival across cohorts. Fourth, as differential expression analysis (Appendix A) revealed differential activation of pathways related to gene expression and transcription in tumors’ γδ T cells, the mechanism by which γδ T cells are related to poor prognosis in HNSCC remains unclear. Finally, given missing data on pathologic factors such as PNI, LVI, ENE, and surgical margin status in TCGA, we were not able to include these factors in the multivariable analysis. Still, our findings underscore the importance of better characterizing the functional role of γδ T cells in HNSCC and understanding whether the presence of γδ T cells may predict the need for alternative therapy or the utility and type of immunotherapy, which are important topics for future investigation.

## 5. Conclusions

CIBERSORT deconvolution of TCGA HNSCC RNA-seq data using the LM22 matrix facilitated reliable deconvolution of 22 immune cell subtypes in 517 tumors, providing a rich and comprehensive encyclopedia of the HNSCC tumor immune microenvironment. Importantly, multivariable survival analysis indicated that, in addition to nodal status, γδ T cells are predictive of poor DFS, even when accounting for a variety of well-established clinicopathologic and genomic features of known significance, including age, smoking status, expression subtype, anatomic subsite, and T stage. Differential gene expression analysis across tumors with and without γδ T cells revealed upregulation of pathways related to gene expression and transcription in tumors with γδ T cells, but the mechanisms behind this novel survival association remain unclear. Still, our findings underscore the need to better understand the role of this rare immune cell subtype in HNSCC and whether the presence of γδ T cells may predict the need for adjuvant therapy.

## Figures and Tables

**Figure 1 cancers-15-04855-f001:**
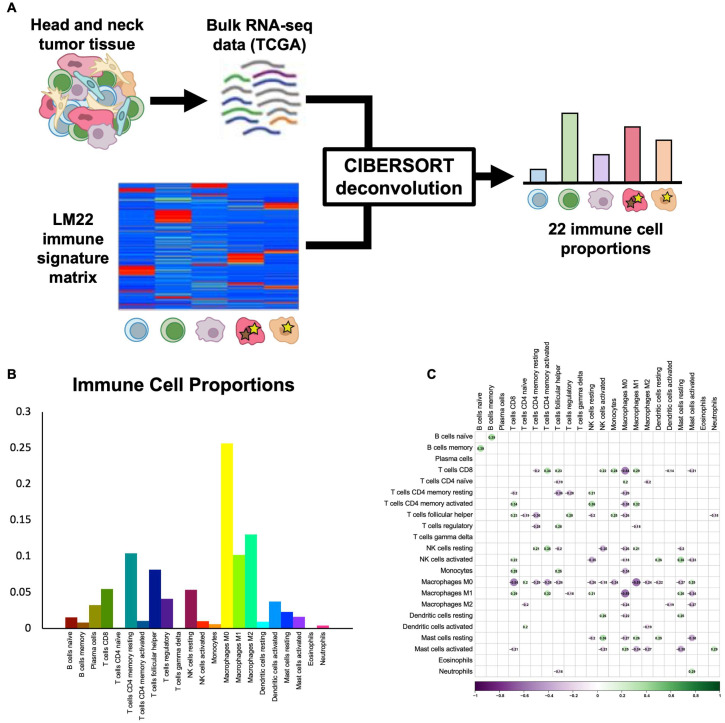
CIBERSORT deconvolution uncovers immune cell proportions. (**A**) Schematic shows workflow of CIBERSORT deconvolution, with TCGA bulk RNA-seq data for HPV-negative HNSCC and the LM22 immune cell signature matrix used as inputs and 22 immune cell proportions returned as outputs. (**B**) Bar plot shows average proportions of 22 immune cell types across TCGA HPV-negative HNSCC tumors. (**C**) Correlation heat map shows Pearson correlations between proportions of immune cell subtypes across samples. No statistically significant correlations were found.

**Figure 2 cancers-15-04855-f002:**
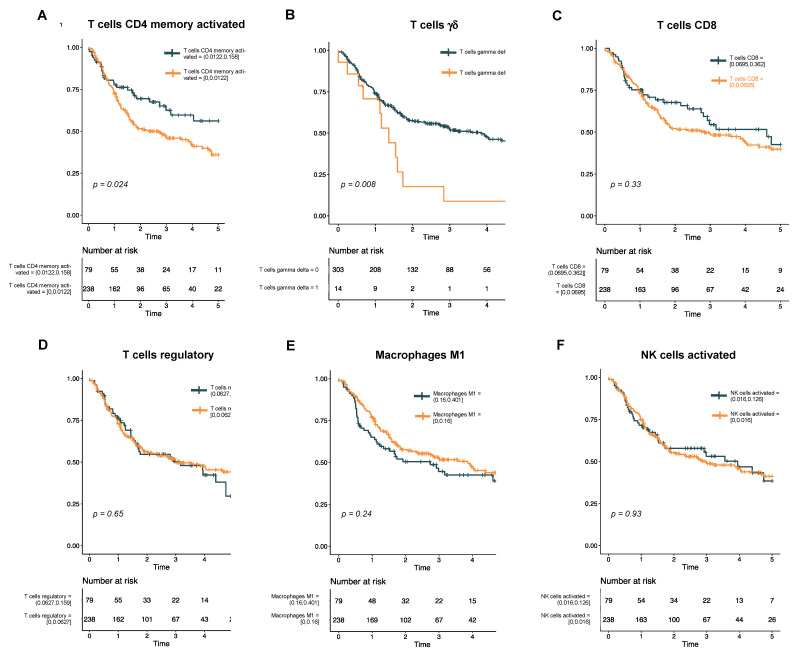
Kaplan−Meier curves show disease-free survival in HPV-negative HNSCC. The cohort is stratified by levels of immune cell subtypes that were significant in the univariate analysis (A, T cells CD4 memory activated; and B, T cells γδ) or have previously been associated with survival in HNSCC (C, T cells CD8; D, T cells regulatory; E, macrophages M1; and F, NK cells activated).

**Table 1 cancers-15-04855-t001:** Multivariable Cox proportional hazards model of factors associated with disease-free survival in HPV-negative HNSCC. Cell types with a significant survival association in the univariate analysis were included in this multivariable analysis, along with other demographic, pathologic, and genomic features typically associated with survival. Perineural (PNI) and lymphovascular (LVI) invasion were not included due to significant missing values. Given the necessity for complete case analysis, patients with missing values for any of the included variables were excluded. Total N = 287 for multivariable analysis.

Factor	HR [95% CI]	*p*-Value
**Age**		
<60	1	
60+	1.28 [0.88–1.85]	0.19
**Smoking**		
Never/former	1	
Current	1.29 [0.88–1.87]	0.19
**Subtype**		
Basal	1	
Atypical	0.75 [0.40–1.41]	0.37
Classical	1.26 [0.73–2.17]	0.41
Mesenchymal	1.28 [0.84–1.94]	0.25
**Subsite**		
Oral cavity	1	
Larynx	0.90 [0.56–1.45]	0.67
Oropharynx HPV-negative/missing	1.28 [0.66–2.49]	0.47
**T stage**		
T1–T2	1	
T3–T4	1.17 [0.80–1.71]	0.43
**N stage**		
N0–N1	1	
N2–N3	1.70 [1.18–2.45]	0.004
**T cells CD4 memory activated**		
[0,0.0112]	1	
(0.0112,0.158]	0.69 [0.44–1.10]	0.12
**T cells γδ**		
Zero	1	
Non-zero	2.13 [1.10–4.13]	0.03

CI, confidence interval.

## Data Availability

All data will be made available upon request. Raw data used as input for immune cell deconvolution and multivariable analysis were downloaded directly from The Cancer Genome Atlas (https://portal.gdc.cancer.gov/, dbGaP study accession phs000178).

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
