# Peer review of "Immune Cell Deconvolution Reveals Possible Association of γδ T Cells with Poor Survival in Head and Neck Squamous Cell Carcinoma"

_cancers, 2023, doi:10.3390/cancers15194855_

Round 1

Reviewer 1 Report

Page 2- Authors didnt mention about the clinical information of the data set being HPV status of remaining non-oropharyngeal data set?

Page 3 - Node positive and node negative category would be more beneficial 

overall staging of HNC should also been included in the analysis as it would give a better relation between immune cell types and disease and for its overall applications.

PNI is an independent factor influencing addition of adjuvant radiation , thus a potential confounder .

Also data on perinodal extension and its margin positivity is missing as these are the major factors influencing overall survival and indication of adjuvant treatment in head and neck cancers.

Also what was the modality of the treatment offered to the patient is also missing as different sites of head and necka cancers have different standard of care !

Result section - 200 patients were lost due to insufficient deconvolution , were any other methods sorted to involve these patients ? As it would significantly strengthen your results.

Absolute number of the specific immune cell authors are concentrating is too small to derive any conclusion from. (0.1%).

Images can be clearer.

Table : 317 patients were included  for final analysis while for multivariate it was 310, a reason for exclusion and how many were gamma delta T cell positive, CD4 activated rested memory cell? while in tables its reduced to 303.

Conclusion need to include the potential factors which were considered in analysis (except PNI and PERINODAL EXTENSION, MARGIN POSITIVITY, ADJUVANT TREATMENT DATA IS NOT INCLUDED)

no issues 

Author Response

  1. Page 2- Authors didnt mention about the clinical information of the data set being HPV status of remaining non-oropharyngeal data set?

Thank you for this comment. HPV status is not available for non-oropharyngeal HNSCC tumors in the TCGA dataset so this variable was unable to be assessed in the entire cohort. However, Bratman et al., JAMA Oncol, 2016 report on HPV status in TCGA HNSCC tumors and found that there were a total of 73 HPV+ tumors, of which the majority (n=53, 71%) were oropharynx tumors, indicating that the prevalence of HPV+ tumors in other head and neck subsites is low and would not enable a statistically relevant stratification of HPV status in tumors of other subsites. In addition, in the clinical management of head and neck cancer, HPV status is considered prognostically relevant only in oropharyngeal tumors (and not in tumors of the oral cavity or larynx), as it has been shown that the presence of HPV DNA in these other subsites does not represent biologically active HPV (Hubbers & Akgul, Virulence, 2015). As a result, we do not believe including this variable would affect the results of the present study.

  1. Page 3 - Node positive and node negative category would be more beneficial. Overall staging of HNC should also been included in the analysis as it would give a better relation between immune cell types and disease and for its overall applications.

Thank you for this comment. As suggested, we performed the multivariable proportional hazards model using N0 vs. N+, and we found a similar result for gamma delta T cells (HR 1.87, 95% CI 1.04-3.39, p = 0.038). In this model, N-stage lost significance as a predictor of survival (HR 1.22, 95% CI 0.95-1.57, p = 0.13). As N-stage is widely recognized as a strong significant predictor of outcomes in HNSCC, we chose to include the binary classification (N0/N1 vs. N2/N3) that highlights this known effect. Overall staging was not included in the analysis because T and N staging were included separately, and the inclusion of overall staging would have been redundant with these two major components of overall stage.

  1. PNI is an independent factor influencing addition of adjuvant radiation , thus a potential confounder. Also data on perinodal extension and its margin positivity is missing as these are the major factors influencing overall survival and indication of adjuvant treatment in head and neck cancers.

We agree that pathologic factors such as PNI, extranodal extension, and positive margins can affect treatment paradigms and thus outcomes. Unfortunately, information on these factors was missing for a significant fraction of patients: PNI was missing for 84 patients, LVI for 90 patients, and ENE (by definition) for all N0 patients. Since the multivariable model requires a complete case analysis (i.e., inclusion of only cases with data available for all variables included), inclusion of these variables would have critically reduced the sample size and limited the ability to draw conclusions. We have included a line about this as a limitation: “Finally, given missing data on pathologic factors such as PNI, LVI, and ENE in TCGA, we were not able to include these factors in the multivariable analysis.”

  1. Also what was the modality of the treatment offered to the patient is also missing as different sites of head and necka cancers have different standard of care !

Thank you for this comment. We agree that a limitation of the use of TCGA data, which is multi-institutional and lacks information on treatment paradigms, is potential variability in treatment paradigms across patients. We have included a line stating this: “Third, as treatment paradigms of patients included in TCGA may vary and are not detailed explicitly, cross-institutional and subsite-related differences may drive differences in survival across cohorts.”

  1. Result section - 200 patients were lost due to insufficient deconvolution , were any other methods sorted to involve these patients ? As it would significantly strengthen your results.

Thank you for this comment. The loss of patients due to insufficient deconvolution is a limitation of the methodology used, and it is not possible to integrate results from an alternate deconvolution strategy for this subset of patients. We have better acknowledged this as a limitation of the work in the discussion section: “Moreover, 200 patients were lost due to insufficient deconvolution, thus limiting the sample size used in the final analyses.”

  1. Absolute number of the specific immune cell authors are concentrating is too small to derive any conclusion from. (0.1%).

Thank you for this comment. We agree that only a small number of samples had gamma delta T cells (14/317, 4.4%) and thus it is difficult to draw definitive conclusions from this work. We instead propose that our work is hypothesis-generating and needs to be validated in further prospective studies. We have included a statement in the discussion clarifying this: “Second, given the rarity of gd T cells, these cells were only detected in 14 patients in our cohort. As a result, these analyses may thus more accurately be seen as hypothesis-generating, and there is a need to reproduce these results in an independent cohort to support clinical utility.”

  1. Images can be clearer.

Thank you for pointing this out. The quality of the images in Figure 2 has been improved, and the yellow text has been changed to black, as requested.

  1. Table : 317 patients were included  for final analysis while for multivariate it was 310, a reason for exclusion and how many were gamma delta T cell positive, CD4 activated rested memory cell? While in tables its reduced to 303.

Thank you for this comment. We apologize for the confusion. We have corrected the number used for complete case multivariate analysis to n=287. Of the 317 patients with reliable deconvolution, patients were excluded due to missing data for one or more of the included variables, given the need for complete case analysis. Of the final 287 patients included in the multivariate analysis, 13 (4.5%) were positive for gamma delta T cells, and 69 (24%) were in the high group for CD4 memory activated T cells.

  1. Conclusion need to include the potential factors which were considered in analysis (except PNI and PERINODAL EXTENSION, MARGIN POSITIVITY, ADJUVANT TREATMENT DATA IS NOT INCLUDED)

We have now included the variables considered in the multivariate analysis in the conclusion: “Importantly, multivariable survival analysis indicated that, in addition to nodal status, gd T cells are predictive of poor DFS, even when accounting for a variety of well-established clinicopathologic and genomic features of known significance, including age, smoking status, expression subtype, anatomic subsite, and T stage.”

Reviewer 2 Report

Anuraag S. Parikh and co-authors presented very interesting work related to the microenvironment in head and neck squamous cell carcinomas (HNSCC). Moreover, we are witnessing a time when immunotherapy is an important modality of treatment for many types of tumors, with the fact that the role of rare immune cell subtypes in many solid tumors is still not sufficiently researched. This is the reason why such research is significant.

My questions (which can be implemented in the discussion) are:

1. Do you know about the presence of this type of T cells in precancerous lesions? And if so, what is the impact on progression to squamous cell carcinoma?

2. HNSCCs have a similar biological behavior, but still if we look at the subregions there are differences. Your cohort consisted of 226 oral cavity, 70 larynx, and 21 oropharyngeal HPV negative tumors. Do you get the same results if you homogenize the group? For example, analysis of laryngeal cancer only?

Additionally, even without these answers, I consider the paper interesting and important for future research, and I congratulate the authors!

Author Response

Anuraag S. Parikh and co-authors presented very interesting work related to the microenvironment in head and neck squamous cell carcinomas (HNSCC). Moreover, we are witnessing a time when immunotherapy is an important modality of treatment for many types of tumors, with the fact that the role of rare immune cell subtypes in many solid tumors is still not sufficiently researched. This is the reason why such research is significant.

We thank the reviewer for his/her enthusiasm for our work!

My questions (which can be implemented in the discussion) are:

  1. Do you know about the presence of this type of T cells in precancerous lesions? And if so, what is the impact on progression to squamous cell carcinoma?

Thank you for this interesting point. To our knowledge, gamma delta T cells have not been studied in head and neck dysplasia, but we agree that delineating the role of this cell type in progression to cancer is a critical next step in understanding the impact of these cells in the head and neck. We have added a statement to the discussion making this point: “They also underscore the importance of specifically understanding the prognostic role of gd T cells in HNSCC, including any potential role in facilitating the progression from dysplasia to carcinoma.”

  1. HNSCCs have a similar biological behavior, but still if we look at the subregions there are differences. Your cohort consisted of 226 oral cavity, 70 larynx, and 21 oropharyngeal HPV negative tumors. Do you get the same results if you homogenize the group? For example, analysis of laryngeal cancer only?

We agree that there are potential differences across subsites within the head and neck, particularly with regard to gene expression, that may lead to differences in immune infiltrate. In the complete case multivariate analysis, there were a total of 207 oral cavity, 62 larynx, and 18 oropharynx HPV-negative tumors, and within the larynx and oropharynx groups, only 1 patient in each was positive for gamma-delta T cells. Given the low incidence of gamma-delta T cells overall, it was not possible to perform a subsite-specific analysis, and the oral cavity predominance of our cohort certainly represents a limitation of this dataset. We have added a statement to the discussion highlighting this limitation: “In addition, given the predominance of oral cavity subsite tumors in TCGA, these results warrant validation in larger cohorts of larynx and oropharynx SCC patients to determine whether these results extend to other subsites”.

Round 2

Reviewer 1 Report

Majority of the issues seems to be resolved with addition/correction by the Author and team except for few.

1. Conclusion includes the following statement "We uncovered that  T cells in the tumor microenvironment were a negative predictor of survival among patients with untreated HPV-negative HNSCC". Thus authors agree that HPV status of the non-oropharyngeal cancers was not known and I agree with the Authors on the non-significant importance of the HPV status in non-oropharyngeal cancers thus categorically involving "HPV- negative HNSCC" should be removed from the conclusion.

2. Not including overall staging would limit its application in real world setting as majority of the literature available are categorised according to overall staging. 

3. Surgical margin status needs to be included to the statement. "“Finally, given missing data on pathologic factors such as PNI, LVI, and ENE in TCGA, we were not able to include these factors in the multivariable analysis.”

Author Response

  1. Conclusion includes the following statement "We uncovered that gd T cells in the tumor microenvironment were a negative predictor of survival among patients with untreated HPV-negative HNSCC". Thus authors agree that HPV status of the non-oropharyngeal cancers was not known and I agree with the Authors on the non-significant importance of the HPV status in non-oropharyngeal cancers thus categorically involving "HPV- negative HNSCC" should be removed from the conclusion.

Thank you for this comment. We have removed “HPV-negative” from the conclusion, both in the abstract and in the main text.

  1. Not including overall staging would limit its application in real world setting as majority of the literature available are categorised according to overall staging. 

Thank you for this comment. Overall stage has now been included in Scheme 2.

  1. Surgical margin status needs to be included to the statement. "“Finally, given missing data on pathologic factors such as PNI, LVI, and ENE in TCGA, we were not able to include these factors in the multivariable analysis.”

Thank you for pointing this out. “Surgical margin status” has now been included in this statement.